# The Origin of Shared Emergent Properties in Discrete Systems

**DOI:** 10.3390/e27060561

**Published:** 2025-05-26

**Authors:** Les Hatton, Greg Warr

**Affiliations:** 1School of Computer Science and Mathematics, Kingston University, London KT1 1LQ, UK; lesh@cantab.net; 2Department of Biochemistry and Molecular Biology, Medical University of South Carolina, Charleston, SC 29425, USA

**Keywords:** emergent properties, Hartley–Shannon information, super-aggregator, CoHSI

## Abstract

Here, we propose that the shared emergent properties reproducibly observed in discrete systems can be explained by a theory that embeds the Conservation of Hartley–Shannon Information (CoHSI) in a statistical mechanics framework. Specific predictions of global properties that represent the most likely equilibrium state should be apparent in all qualifying systems, regardless of provenance. We demonstrate that these predictions of emergent global properties hold true in systems as disparate as collections of software written in the programming language C and collections of proteins. The implication is that the emergence of such shared properties is not driven by any specific local mechanism as the systems are so different. This raises the interesting prospect that important properties of biological systems (exemplified here by the length and multiplicity distributions of proteins) have little, if anything, to do with natural selection. Similarly, the size distribution of components and the frequency of tokens observed in computer software in C emerge as the most likely states, and are thus properties that are divorced from human agency, regardless of functionality.

## 1. Introduction

The goal of this paper is to demonstrate, through both proof and measurement, that discrete systems of completely different provenances can and do share the same emergent properties. By a discrete system, we mean one assembled from components that each comprise discrete, distinguishable pieces (c.f. Appendix A). We will see that such systems share properties which do *not* derive from any intrinsic meaning the pieces might have. Furthermore, these properties appear alongside, and are entirely co-existent with, any properties that *do* derive from any intrinsic meaning the pieces might have.

To consolidate this, consider Figure 1. Without further insight into what the string of coloured beads represents, this is equally likely to be a primitive decorative wrist band or to represent a string of amino acids in a protein, or it might be a representation of a digital message, with each uniquely coloured bead corresponding to a different symbol. However, it also has mathematical properties deriving *solely* from the fact that the different coloured beads and their order are distinguishable. These latter properties should be present, therefore, whether the string represents a protein, a bracelet, or a string of bases in a DNA molecule. They are shared properties.

We will argue that all discrete systems share this duality and we will demonstrate this theoretically and then on real systems in biology and in software.

We should note that this is not an unrealistic objective, as the reliable emergence of global patterns is well-documented from many sources and experiments; for example, in the astonishingly ubiquitous Zipf’s law [1,2,3].

There is something slightly unnerving about Zipf’s law, generally considered to be an empirically observed power-law. We like to think as authors that we have entire control over what we write, but George Zipf observed nearly a century ago that if word frequencies were extracted from written text, independently of their language or subject matter, and arranged in rank order, they closely follow an almost identical power-law with a slope which Zipf’s data suggested was “approximately” unity [4]. There are few more emphatic examples of hidden clockwork than to analyse a newly written book on any subject in any language only to find that the word frequencies obey a distribution which has nothing to do with the author(s) use of language or their subject and of which the author was completely unaware. The words will typically differ in rank order from text to text but their frequency vs. rank order distribution will follow Zipf’s Law. Cherry, in [5], describes a proof using Mandelbrot for texts but the reason for the ubiquity of Zipf’s Law in systems as diverse as wealth distribution [6] or the size of lunar craters [7] were unknown until 2019, when a form of Zipf’s Law was shown to be the equilibrium state of all discrete systems of a particular kind the authors called *homogeneous* systems [8]. Homogeneous systems are those for which the constituent discrete pieces are *unordered*. A closely related and equally ubiquitous kind of system in which the discrete pieces are *ordered*, termed a *heterogeneous* system, is also identified in [8]. Heterogeneous systems are also dominated by power-law behaviour but are distinctly different from homogeneous systems. We consider both here and, given its ubiquity, will interpret Zipf’s law as generic power-law behaviour where the slope may differ significantly from the originally documented “approximate” unity. As we discuss later, this has become common parlance.

The key to deriving a proof of Zipf’s Law and to understanding the ubiquitous emergence of reproducible patterns in discrete systems, whether pieces are ordered or unordered, is based around the inherent ambiguity of Figure 1. In this case, the beads are the abstract representation of the pieces of any ordered discrete system. If there is to be any chance of explaining such shared emergent properties, then the underlying theory must be independent of any intrinsic meaning of the discrete pieces other than that they are distinguishable. The argument we develop here begins with a consideration of the most likely state of a system and the classical concept of entropy.

### 1.1. Entropy as an Emergent Property

Readers of this journal will be well aware of the perceived complexities of entropy but it will be useful to the discussion which follows to revisit Boltzmann’s masterful development of the concept. Entropy has prosaic origins in the efforts of 19th century engineers such as Carnot, Clausius and Kelvin, who developed what became Classical Thermodynamics and the study of heat engines, notably the steam engine of the era. In essence, entropy is the accumulation of large numbers of infinitesimal irreversible changes as a heat engine carries out its work. It is an emergent property, as pointed out by [9], and is easily calculable as essentially the heat transferred in a system in which no work is done divided by the absolute temperature at which it is transferred, as has been calculated routinely by chemists and engineers for more than a century. It is emergent in the sense that we cannot identify it until we look at the system as a whole as it evolves over time. It is most definitely not a local mechanism.

Entropy is at the heart of the second law of thermodynamics and is typically given the symbol *S*. For a closed system, changes in *S* are non-negative and it is a mistake to think that life, or indeed any other process, can alter this fact [10]. It is intimately bound to the Conservation of Energy, which itself is intimately bound to one of the great symmetries in the universe, the invariance of physical experiments in time, [11].

Roughly concurrently with this work, the great Austrian physicist Ludwig Boltzmann was trying to reconcile an atomistic view with classical thermodynamics, a wholly macroscopic viewpoint. In essence, he considered a collection of molecules, each of which could take one of a finite set of *M* energy levels, and was interested in how many ways those molecules could be distributed amongst those energy levels subject to the twin constraints of a fixed number of molecules *T* and a fixed total energy *E*, where T,E are given by(1)$$T=\sum_{i=1}^{M}t_{i}$$ and (2)$$E=\sum_{i=1}^{M}t_{i}\epsilon_{i}$$

In this nomenclature, the $i^{th}$ energy level $\epsilon_{i}$ is occupied by $t_{i}$ molecules.

Boltzmann then asked what the most likely distribution of energy would be amongst the *M* levels in all possible systems with that *specified T* and *E*. He approached this by first writing down the total number of ways in which the $t_{i}$ could be arranged in each of their *M* energy levels, initially assuming that the molecules were otherwise indistinguishable.(3)$$\begin{array}{c} \Omega =\frac{T!}{\prod_{i=1}^{M}\left(t_{i}!\right)} \end{array}$$

At this point, using the work of the French mathematician Lagrange, he was able to write down the total number of ways in which this could be achieved, combining (Equation 3) subject to the constraints of Equations (Equation 1) and (Equation 2) as(4)$$\begin{array}{c} \ln\left(\Omega^{\prime }\right)=\ln\left(\frac{T!}{\prod_{i=1}^{M}\left(t_{i}!\right)}\right)+\alpha \left[T-\sum_{i=1}^{M}t_{i}\right]]+\beta \left[E-\sum_{i=1}^{M}t_{i}\epsilon_{i}\right]] \end{array}$$

Here, α and β are *Lagrange undetermined multipliers*. Note that Boltzmann combined the log of the Ω function (to simplify the factorials) with the two constraints on size and energy. First, simplify this as(5)$$\begin{array}{c} \ln\left(\Omega^{\prime }\right)=\ln\left(T!\right)-\sum_{i=1}^{M}\ln\left(t_{i}!\right)+\alpha \left[T-\sum_{i=1}^{M}t_{i}\right]]+\beta \left[E-\sum_{i=1}^{M}t_{i}\epsilon_{i}\right]] \end{array}$$

We use $\Omega^{\prime }$ to emphasise the presence of the two constraints. Boltzmann then simplified this further using the well-known Stirling approximation:(6)$$\begin{array}{c} \ln(N!)\approx N\ln N-N \end{array}$$

Using (Equation 6) in (Equation 5) and varying this with respect to the $t_{i}$, keeping the $\epsilon_{i}$ constant, to find the distribution that maximises $\ln(\Omega^{\prime })$ by setting $\delta (\ln\left(\Omega^{\prime }\right))=0$, he arrived at the following:(7)$$\begin{array}{c} \sum_{i=1}^{M}\delta t_{i}\left[\ln\left(t_{i}\right)+\alpha +\beta \epsilon_{i}\right]=0 \end{array}$$

Finally, *since the $\delta t_{i}$ are completely arbitrary*, the only way this set of equations can always sum to zero is if the inner parentheses are identically zero *for any i*, giving(8)$$\begin{array}{c} \ln\left(t_{i}\right)+\alpha +\beta \epsilon_{i}=0 \end{array}$$

From this, it trivially follows that the overwhelmingly most likely distribution of the particles amongst the energy levels of all possible systems with the same total size and energy is(9)$$\begin{array}{c} t_{i}=e^{-\alpha -\beta \epsilon_{i}}=Ae^{-\beta \epsilon_{i}} \end{array}$$

The factor $A=e^{-\alpha }$ is simply a normalisation factor. Another way of looking at (Equation 9) is that it is the equilibrium state. Note that this formulation also presents this as the overwhelmingly most likely state. Other states are possible but are very unlikely and could be interpreted as departures from the equilibrium state. We will say a little more about overwhelmingly likely states later.

Equations (Equation 1)–(Equation 9) are a completely standard development, as described by [12,13].

However, (Equation 9) is not closed since β is unspecified as a Lagrange undetermined parameter. With a revolutionary flash of insight, Boltzmann guessed that the *S* of classical entropy obeyed the following:(10)$$S=k.ln(\Omega^{\prime })$$
where *k* is a constant. This closes the system, marrying the two great theories of Statistical Mechanics and of Classical Thermodynamics together because it turns out that Equation (Equation 10) implies $\beta =1/(k_{B}Temp)$ (and vice versa; see p. 19–20 of [13]), where $k_{B}$ is now known as Boltzmann’s constant and Temp is the absolute temperature.

This is an important point for what follows. Boltzmann derived an infinite class of solutions but guessed that a subset of this family corresponded to classical thermodynamics, thus fixing β to the physical concept of temperature and, as can be proven, establishing Equation (Equation 10).

Now, recall that *S* is the classical entropy, inextricably bound to the conservation of energy and non-decreasing in any closed system. However, Boltzmann provided us with an infinity of other possibilities when β is *not* tied to the absolute temperature. Under this umbrella lies what is now known as *Information Entropy*.

### 1.2. Information Entropy and the Development of CoHSI Theory

It is important that information entropy not be confused with classical entropy. It has the same mathematical form as derived by Boltzmann but β remains undetermined and it is not, therefore, tied to classical entropy and, by implication, energy.

We can at last begin to engage with the real subject of this paper: the search for the origin of shared emergent properties in discrete systems. The theoretical development of the Conservation of Hartley–Shannon Information (CoHSI) that follows first appeared as [8] but, for convenience, a brief summary is provided here.

Our starting point is to realise that Boltzmann’s methodology is remarkably flexible and we can constrain anything in Equation (Equation 4) for which the constraint is additive (as is the case for both energy and information). However, we must achieve this in a manner which is independent of the local mechanisms of any particular system if we are to find an emergent property shared by very disparate systems.

We begin by defining(11)$$I=\sum_{i=1}^{M}t_{i}I_{i}$$
where *I* is the total information content of the system and $I_{i}$ is the information content of each of the $t_{i}$ elements of the $i^{th}$ level, leaving aside what this might look like for now. We therefore rewrite Equation (Equation 4) in a variational form as(12)$$\begin{array}{c} \ln\left(\Omega^{\prime }\right)=\ln\left(\frac{T!}{\prod_{i=1}^{M}\left(t_{i}!\right)}\right)+\alpha \left[T-\sum_{i=1}^{M}t_{i}\right]]+\beta \left[I-\sum_{i=1}^{M}t_{i}I_{i}\right]] \end{array}$$
where *T*, the total size of the system, is defined as Equation (Equation 1).

Before continuing with our development, we should note that merging some form of information with the methodology of statistical mechanics is not new. Versions of this include the work of [14], who used this methodology and Shannon entropy to show that observations of the properties of data were actually synonymous with particular types of statistical distribution. For example, if the average value of the observations summarises all of the information available in the data about the distribution, then an exponential distribution is the most likely outcome. These are interesting insights but are difficult to apply in practice and are tangential to this discussion.

#### 1.2.1. Why the Conservation of Information?

Before considering the two kinds of system which naturally emerge in CoHSI theory, it is reasonable to ask whether we are restricting ourselves only to systems which conserve information. After all, it might be expected that, as life evolves, the total amount of information will change. Does this in any way derail CoHSI?

In fact, we are not thus restricting ourselves as we are only considering the Conservation constraints in a very specific way relating to how the asymptotic distribution in Boltzmann’s theory emerges.

The variational question we are asking (if considering energy) is as follows:


*Given all possible systems of the same fixed size and the same fixed total energy, what is the most likely distribution of energy amongst the various discrete pieces, (levels, molecules, etc.)?*


Whereas, if considering information, we are asking the following:


*Given all possible systems of the same fixed size and the same fixed total information, what is the most likely distribution of information amongst the various discrete pieces, (tokens, etc.)?*


In both cases, we are envisaging an ergodic ensemble of all possible systems with the same constraints, and in both cases we could indeed imagine either further energy being injected into the system or total information increasing, but the beauty of Boltzmann’s vision is that the distributions are scale-independent. Thus in a gas, if you increase the total energy by, for example, heating it, you still have a Maxwell–Boltzmann distribution of velocities. By analogy, in an information system in which the total information is changed, as in a living system, you will still have the CoHSI distributions because whether the constraint is on energy or information, the same underlying scale-invariant statistical mechanics methodology is used. The mathematics clearly implies this, and it is sufficiently important that we test and confirm this scale-independence later for two very large datasets of entirely different provenances, which are considered individually as they grow by at least a factor of 10.

#### 1.2.2. Ordered or Heterogeneous Systems

“Information” is a much exercised word in the 21st century and we will use the simplest and arguably the most pure definition, provided by Ralph Hartley in 1928 [15], which has all the properties we seek. It as additive but the only thing it requires is that the discrete pieces are distinguishable, eschewing any intrinsic meaning. To understand Hartley’s definition of information, consider a collection of strings of varying lengths, of the kind shown in Figure 1. Each string has an alphabet of colours from which the string can be constructed; more than one bead of the same colour is allowed. The alphabet of each string will typically be a subset of the combined alphabet of all strings.

Hartley was dealing with the problem of sending a message made of distinguishable symbols from an alphabet down a communication line. Each message looks similar to Figure 1: a string of distinguishable symbols. He argued that, from the point of view of the communication line, any message is possible, so any of the individual symbols must be equally likely. Furthermore, there can be no implied meaning attached to a symbol. He therefore defined information as follows [5,15]:


*The information content of a message is the log of the number of ways the symbols in the message can be arranged, specifically excluding any intrinsic meaning so that all symbols are equally likely.*


The information content of a message of N symbols drawn from a possible unique alphabet of P symbols, where any symbol is equally likely, is therefore(13)$$I=\log\left(P\times P\times \ldots \times P\right)=\log\left(P^{N}\right)=N.\log\left(P\right)$$

Although Shannon built on this for symbols with non-equal probabilities [16,17], we will refer to Equation (Equation 13) as Hartley–Shannon and will use this as our measure $I_{i}$ of the information content of the $i^{th}$ component. The only requirement is that the discrete pieces (symbols in Hartley’s wording and tokens in ours) are distinguishable and equally probable but otherwise without any intrinsic meaning. This is just the property we need. We can thus use a model of strings like Figure 1 to mean proteins (with the beads corresponding to ordered amino acids) or software (with the beads corresponding to ordered programming language tokens), or indeed any similar system, and if our theory is correct, we should find that they have shared emergent properties.

#### 1.2.3. Unordered Systems

We mentioned above that another kind of system, the unordered or *homogeneous* system, also emerges from CoHSI theory. Homogeneous systems, exemplified by Figure 2, encompass those systems in which Zipf’s law has historically been observed. For example, we could assign a different bead colour to each unique word in a text and, as we encounter each word going through a book, we just place a bead of the appropriate colour in the box assigned to that colour. Note that no ordering is implied. The purple beads in Figure 2 might correspond to occurrences of the word “the”, but no ordering is implied in how these were encountered.

When this information model is placed in (Equation 12), a form of Zipf’s law emerges as the predicted equilibrium state; this is a power-law distribution with a characteristic droop in the tail. Any system that can be categorised as in Figure 2 will also be overwhelmingly likely to exhibit this Zipfian behaviour, since the beads have no intrinsic meaning. We argue that this is not a mystery; it is inevitable.

### 1.3. On Equilibrium

It is interesting to contrast the two functionals: Equations (Equation 5), the energy formulation, and (Equation 12), the information formulation. We left the same undetermined multipliers in there as α,β; they will have different values, but how different, and how important are these differences? In both cases, alpha acts only to normalise. We focus on β.

In the case of conserved energy, we know from classical thermodynamics that(14)$$\beta =\frac{1}{k_{B}\left(Temp\right)}$$

The first thing we notice in the energy case is that β has the dimensions of $J^{-1}$, where *J* is Joules and, for typical ranges of temperatures on earth, $\beta ∼10^{21}\;J^{-1}$. This would suggest that the β term of Equation (Equation 5)(15)$$\beta \left[E-\sum_{i=1}^{M}t_{i}\epsilon_{i}\right]$$

Would dominate unless the differences between *E* and $\sum_{i=1}^{M}t_{i}\epsilon_{i}$ relative to the total energy were infinitesimal, which is reassuring.

However, in the case of conserved information, there is a marked contrast wherein the typically observed values of β in power-law systems are less than 4 [2]. In this case, given the similarity of Equations (Equation 5) and (Equation 12), larger departures from the conserved information might be tolerated to keep the separate terms balanced in magnitude in Equation (Equation 12).

This is an important point. CoHSI is not a straitjacket and, from the argument above, is a relatively weak constraint, in contrast to the conservation of energy. This means that we are quite likely to see discrete systems depart from the equilibrium. We would argue that this is not a breakdown in CoHSI; in fact, it is to be expected. In other words, when we see minor departures from the equilibrium distributions, such as those that we will describe below in Figure 3A,B, we should also consider the possibility of a power-law distribution, which is simply out of equilibrium.

## 2. Materials and Methods

### 2.1. Databases

#### 2.1.1. Biological Datasets

The latest version of the full TrEMBL protein database TrEMBL 24-06 is to be found at https://ftp.uniprot.org/pub/databases/uniprot/knowledgebase/complete/uniprot_trembl.dat.gz (accessed on 8 January 2025), and versions taken from the past ten years of this dataset—versions 15-07, 17-03, 19-04, and 22-02—can be found at https://ftp.uniprot.org/pub/databases/uniprot/previous_releases/ (accessed on 8 January 2025). The nomenclature is YY-NN, where NN is the release number in year YY. Since July 2015, the total number of proteins has increased by almost a factor of 10 to its current level of something over 251 million proteins.

#### 2.1.2. Software Datasets

The 83.5 million lines of open source code used here include the Linux kernel, the BSD (Berkeley Software Distribution) release, X11R7, the GNU collection of utilities, and major open source projects such as PHP, Gimp, Perl, postfix, Apache, the GNU compiler, Gtk, ImageMagick, Ingres, the KDE libraries, Mysql, Postgresql, R, realplayer, tcltk, and numerous smaller projects. All are freely downloadable, mostly from Github, although Linux source distributions are available in any Linux distribution. These were accessed directly from the Linux kernel archive as follows:


    $ git clone \
       git://git.kernel.org/pub/scm/linux/kernel/git/stable/linux.git
    $ cd linux
    $ git checkout ...


The software analysed in this paper was exclusively written in the programming language C (ISO C90 or ISO C99, https://iso-9899.info/wiki/The_Standard (accessed 22 May 2025) with the majority compiled by the GNU C compiler), the primary reason for this being one of its availability and easily accessible open source software; the majority is written in C [18]. A secondary reason is purely logistical. Parsing protein strings is relatively trivial in computing terms, requiring only relatively simple lexers, which we wrote in perl, but parsing programming languages requires the development of what is essentially a compiler front-end for each language to perform lexical and, in some cases, syntactic analysis sufficient to unambiguously recognise components. These were written in C using open source compiler support tools such as *flex* [19,20]. Further complexity exists in the case of object-oriented languages and tokens. It is an interesting philosophical point, but beyond the scope of this paper, to compare protein parsing, which is lexically trivial but with syntactic and semantic rules of great and mostly unexplored complexity, with software parsing, which is lexically complicated but with relatively simple syntactic and semantic rules.

### 2.2. Computational Methods

The complete software used to compute all diagrams and statistical results in this paper is available for download from https://leshatton.org/HattonWarr_SharedProps_Apr2025.html (accessed 22 May 2025). The datasets used are all publicly available. We should note that repeating these results is not a trivial undertaking. The latest TrEMBL protein database used, version 24-06, is around 192GB when compressed, which would expand to around 1TB uncompressed. This has grown by over 20% in just two years. Processing it required a substantial computer. We used an AMD Ryzen 9 32-thread 3.4GHz machine with 32GB of main memory and twin 8TB high speed discs. Even so, and with the parallelisation we built into the software to keep all 32 threads fully occupied for some parts of the computation, the overall compute time was still several hours.

### 2.3. The Canonical CoHSI Distributions

The overwhelmingly most likely solution of Equation (Equation 12) is given in [8] as(16)$$\log t_{i}+\frac{1+8t_{i}+24t_{i}^{2}}{6(t_{i}+4t_{i}^{2}+8t_{i}^{3})}=-\alpha -\beta \left(\frac{dI_{i}}{dt_{i}}\right),$$

If Ramanujan’s factorial approximation is used [21].

For the heterogeneous case, we can calculate the Hartley–Shannon information of each string $I_{i}$ as described in the open access original reference [8] and substitute this in Equation (Equation 16). The solution is a prediction of the length distribution of the strings which looks like Figure 3A.

For the homogeneous case, we calculate the Hartley–Shannon information of the set of boxes of Figure 2 [8] and substitute this in Equation (Equation 16). The result is shown in Figure 3B.

To bridge the gap with the original reference, [8], we provided more details of both derivations in Appendix A, but we strongly recommend consulting the original reference to deepen your understanding.

Note that in both heterogeneous and homogeneous cases, although both solutions are dominated by power-law behaviour, each differs from the precise Zipfian behaviour in its own way. These predicted differences will be observed. Ubiquitous Zipf(ian) behaviour is really ubiquitous Zipf-*like* behaviour; the power-law is eye-catching but is not the only feature of note.

In the results section, we will demonstrate the following:**D1** Shared heterogeneous behaviour between the current latest version of the protein database TrEMBL 24-06 at https://uniprot.org (accessed 22 May 2025) and a set of open source software written in C in the Linux and BSD distributions.**D2** Scale-independent heterogeneous behaviour in the last 10 years of versions of the TrEMBL protein database and in 12 versions of the Linux kernel distribution in the last 15 years (versions 2.6.11 - 6.9). This is written in C.**D3** Shared homogeneous behaviour between the current latest version of the protein database TrEMBL 24-06 at https://uniprot.org (accessed 22 May 2025) and a set of open source software written in C in the Linux and BSD distributions.**D4** Scale-independent homogeneous behaviour in the last 10 years of the TrEMBL protein database and 12 versions of the Linux kernel distribution in the last 15 years (versions 2.6.11 - 6.9). This is written in C.

These will bear out the predictions that (a) shared emergent properties do indeed co-exist in both heterogeneous and homogeneous datasets of very different provenances, as is predicted by CoHSI theory, and (b) that they are scale-independent.

## 3. Results

For our biological dataset, we will use the latest version of the full TrEMBL protein database TrEMBL 24-06, as detailed in the Materials and Methods.

### 3.1. Item D1

The pdf (probability distribution function) and the ccdf (cumulative complementary distribution function) of the length distribution of all approximately 251 million proteins measured in the amino acids is shown in Figure 4A and Figure 4B respectively. Before comparing these with the equivalent distributions for the function lengths measured in the programming language tokens, we must first demonstrate that these results are statistically consistent with the predictions of CoHSI: that heterogeneous systems have a sharp peak followed by an extremely precise power-law. The peak in Figure 4A is obvious but assessing the tail for power-law behaviour is a more protracted process.

It is all too easy to assert power-law behaviour based on the most obvious property—that of a straight line on a log-log plot—as shown in Figure 4B. However, following [22], this is only a necessary condition. For sufficiency, a more detailed check, such as that afforded by the Clauset–Gillespie procedure poweRlaw() in R, is required [23].

Applying both necessary and sufficiency criteria for the power-laws to Figure 4B in the region of 400 to 20,000 amino acids (essentially the region after the peak in the pdf of Figure 4A leads to the results shown in Table 1.

Table 1 clearly shows the predictions of CoHSI. The length distribution of the proteins of Trembl release 24-06 is dominated by a power-law (just as it was in previous releases; [24]). Note that any value of *p* greater than 0.1 in the Clauset–Gillespie test is enough that power-law behaviour cannot be rejected.

We should say a little more about this as, in statistics, it is not unusual to compare the goodness of fit of different probability distributions. In the case of CoHSI, however, because we have an underlying theory and we know that CoHSI is not a straitjacket, it is sufficient to show that the power-law behaviour is not rejected. This is irrelevant if there is a better-fitting alternative distribution: even if a power-law is rejected on a particular dataset by the Clauset–Gillespie test, CoHSI theory can accommodate this by virtue of the fact that it is not a straitjacket. Rather, we can reasonably consider the observed departures from CoHSI to be fluctuations around the equilibrium CoHSI state provided that these are the exception rather than the rule.

So, do we see the same behaviour in software? Are these properties shared as CoHSI predicts? The equivalent pdf and ccdf are shown in Figure 5A,B. The population of open systems written in C used to test the model described here is smaller, totalling some 83.5 million lines, containing almost 500 million tokens.

Again, as shown in Table 2, both necessary and sufficiency tests for power-law behaviour reveal emphatic support for the predictions of CoHSI.

The predictions of CoHSI with respect to function lengths were assessed in seven other programming languages, which obtained similar results even though the quantity of available software for analysis was significantly reduced for languages other than C [25].

Thus, Item D1 has been clearly demonstrated in two systems of completely different provenance (proteins and computer software) that have no mechanisms in common. Both systems exhibit the same distribution, although with different values of slope β.

### 3.2. Item D2

We are seeking here to demonstrate the scale-independence of CoHSI (guaranteed by the absence of *T* in Equation (Equation 16)). We approach this by comparing the length distributions for the totality of proteins contained in five sequential releases of the TrEMBL database, which represent an approximately 10-fold increase in the number of proteins. Consider Figure 6, in which we compare the protein length distributions (as ccdf) for TrEMBL releases 15-07, 17-03, 19-04, 22-02, and 24-06. The scale-independence is obvious. Indeed, another feature of CoHSI solutions, that the maximum protein length is essentially determined by the total number of proteins in the database at any one time, is of considerable interest in itself. Table 3 shows the increase in maximum protein length with the TrEMBL version.

These maximum lengths are surprising. The average protein length is only of the order of 400 or so amino acids (depending on the domain of life). In the simplest case, each position can be one of twenty amino acids, so even an average protein has a staggering number $∼20^{400}$ of possible sequences. Why, then, does nature produce proteins 100 times longer, which require a substantial expenditure of metabolic energy? CoHSI provides a compelling explanation. The scale-independent rise in the total size of the population of proteins as life evolved guarantees the emergence of ever longer proteins to satisfy the CoHSI equilibrium, and thus provides novel opportunities for the evolution of life forms. This has interesting implications. For example, we can consider the protein titin, which is essential to movement because of its multiple roles in muscle function. Titin is also very long (between 27,000 and 35,000 amino acids), which raises the following question: would muscles and the movement they enable have evolved without the role of CoHSI in generating, purely probabilistically, long proteins such as titins?

The hump in the tail of Figure 4B is also of considerable interest. As late as the 18-02 release, the longest known protein was less than 37,000 amino acids long. By the 24-06 release, there were 21 known proteins longer than this. This suggests that, in the relatively near future, the current longest known protein of 45,354 amino acids will be replaced by a longer protein as more and more sequences are discovered and the system tends towards equilibrium.

It is interesting to note in passing that the slope of the power-law in the protein database Figure 6 is almost twice that of the software Figure 7 in spite of the protein database being much larger than the software system used. We will return to this later.

### 3.3. Item D3

Here, we illustrate the commonality of behaviour (in terms of CoHSI homogeneous distributions) in two completely distinct systems: the frequency of multiplicity in the evolution of proteins and the frequency of token occurrence in a large software package. By protein multiplicity, we mean the number of species in which given proteins occur that are identical in both the number and sequence of amino acids. Protein multiplicity is an outcome of evolution in which protein-encoding genes are transferred from organism to organism either vertically (from parent to offspring, as in humans) or horizontally (as in the plasmid-based transfer of antibiotic-resistance genes between different bacterial species) [26]. If we categorise proteins by their multiplicity, we can count the numbers of proteins that share the same multiplicity, exactly fulfilling the requirements of a CoHSI homogeneous system. Thus, CoHSI predicts a power-law distribution of multiplicity, with a droop in the tail in the lowest-populated bins. Combining data on multiplicity across all three domains of life (eukaryota, archaea, and bacteria) and the viruses, a precise Zipfian distribution was first observed by [26] directly as a result of the CoHSI prediction. If we examine the protein multiplicity for the latest version of TrEMBL, version 24-06, in the same manner, we obtain the plot shown in Figure 8, which provides an astonishingly precise power-law across almost four decades.

Applying both necessary and sufficiency criteria for power-laws to Figure 8 in the region of multiplicities from 1 to 8000 gives the following (Table 4):

Once again, an analysis of Table 4 strongly supports the predictions of CoHSI.

We can now compare the behaviour of protein multiplicities in the evolution of life with another homogeneous dataset of completely different provenance: token frequency in a set of software. We use version 6.9 of the Linux kernel, as this is easy to access. These are plotted as token frequency *versus* rank in the usual way on log-log scales, as shown in Figure 9.

This result shows the distribution expected with Zipf’s Law, i.e., one dominated by a power-law. Applying both the necessary and sufficiency criteria for power-laws to Figure 9 in the region from 1 to 5000 unique tokens gives the following (Table 5):

Item D3 has therefore been demonstrated here in two systems of completely different provenances (computer software written in C and the multiplicity of proteins that arose over the course of evolution) with no conceivable mechanisms in common. With the exception of some concavity in the least-populated bins in Figure 9, both systems exhibit the same CoHSI-predicted Zipfian distribution (i.e., a power-law with a droop in the tail) to a high degree of precision, although once again with different values of slope β.

### 3.4. Item D4

Finally, we demonstrate scale-independent behaviour in the multiplicity of proteins measured over a period of 17 years in which the size of the dataset increased by a factor of approximately 100. The multiplicity of proteins was calculated for each of the seven TrEMBL datasets in a manner identical to that used to generate Figure 8. These data are plotted in Figure 10 and illustrate, in a visually compelling manner, the scale independence of the slope of the multiplicity distribution as the size of the database increases by two orders of magnitude.

We compare this with the frequency of the tokens in twelve releases of the Linux kernel stretching back some 15 years, as shown in Figure 11, which also exhibit scale-independence as this software system grew by a factor or around 10 in this period.

## 4. Discussion

The results presented here illustrate, using data taken from systems of very different provenances, that, when categorised appropriately as heterogeneous (ordered) or homogeneous (unordered) discrete systems, the distributions predicted by CoHSI theory are observed. These distributions are dominated (but not completely) by precise power-laws. We interpret the emergence of these reproducibly observable patterns as indicative of two features of discrete systems, as follows:The simultaneous existence of two sets of properties: one set pertaining to when intrinsic meaning is considered and one set pertaining to when intrinsic meaning is irrelevant and therefore shared amongst disparate systems.Discrete systems tend towards the overwhelmingly most likely equilibrium state defined by CoHSI theory, with the caveat that departures from the equilibrium are permitted by the theory. This latter point can be observed in the minor bumps and bulges that are observable in many of the datasets presented here.

If the argument presented in this paper is correct, it has significant implications, particularly for our understanding of life and our interpretations of evolutionary histories.

### Life’s Emergent Properties: The Most Likely Outcome of Evolution

When we look at the life-forms that are in, on, and around us worldwide, their diversity is dazzling. Estimates suggest that their total numbers are astronomical: there may be a trillion microbial species, perhaps a billion species of virus, and nearly 9 million eukaryotic species [27,28,29]. This great diversity of organisms is the outcome of at least 3.5 billion years of evolution; thus, biologists, whatever their field, are inevitably engaged in studying, at some level, (1) the product of evolution (e.g., how organisms work, interact, and cause disease?) or (2) the mechanisms of evolution, in which Darwin’s Theory of Evolution by Natural Selection [30], updated in light of modern genetics and molecular biology, along with concepts of drift (Kimura’s Neutral Theory), constitute the generally accepted paradigm [26,31,32,33]).

However, across the whole of biology, there is also widespread evidence for the emergence of regular, reproducible patterns in the form of robust statistical distributions. The importance of these emergent patterns is that they provide theoretical difficulties for evolutionary theory, in that reproducibly emergent patterns are, ipso facto, properties that are not subject to selection [34]. When biologists consider the details of modern life-forms, such as the fine structure of the genome and the proteins it encodes, as distinct from their evolutionary history, we see once again the widespread emergence of reproducible patterns that are difficult, if not impossible, to explain using reductionist, mechanistic approaches.

This difficulty in explaining the origin of emergent properties in biological systems has led to a silent consensus of denial that there could be underlying physical principles or laws that are responsible for the reproducible global patterns in living systems. A typical argument is to claim that biology is too diverse and too complicated for any simple principle to be reproducibly manifest (for example, as a Zipfian distribution) across so many systems of utterly different characters. How could a single principle operate equivalently on the genome [35,36,37], on proteins [8,26,38,39], on organism abundance [40] and physiology [41], on the timing and magnitude of infectious disease outbreaks [42,43,44,45], and at the level of landscape ecology [46,47], and be evident in the fossil record [48,49,50]? The answer, from the perspective that there are no laws or principles operative in the evolution of life, is that each biological system must have its own unique set of mechanisms that all coincidentally generate the same emergent properties. We argue here that the widespread and implicit assumption that there are no laws or principles that govern the evolution of life has led to emergent properties in biology being underemphasised and underappreciated regarding the insight they provide into the natural world and its evolution. The particular set of emergent properties that we focused on here and referenced above in this section are the homogeneous and heterogeneous distributions predicted by CoHSI theory, with which life is replete, and that, we argue, have particular implications for our understanding of the evolution of life on earth.

## 5. Conclusions

The theory (Conservation of Hartley–Shannon Information or CoHSI) predicts that two canonical power-law distributions dominated by power-law behaviour simply represent the most likely equilibrium state of discrete systems, regardless of provenance. These are the *heterogeneous* distribution predicted for ordered systems and the *homogeneous* or Zipfian distribution predicted in systems that are unordered. If this is correct, then we should be able to demonstrate the shared presence of these predicted properties in unrelated discrete systems.

Here, we demonstrate the accuracy of the predictions of CoHSI in collections of open source software written in C and of proteins (accessed through the TrEMBL protein database at https://uniprot.org, accessed 22 May 2025). In these two systems, we compared the properties of (1) the length distributions of proteins and of software functions, (2) the frequencies of protein multiplicity and of token use in software, and (3) scale-independence. These two systems share no intrinsic meaning in the vocabulary of the discrete pieces of which they are composed, i.e., amino acids in the case of proteins and programming language tokens in the case of open source software. However, in both systems, the analyses strongly support the predictions of CoHSI theory that there are shared emergent properties, such that the following hold:Heterogeneous behaviour, manifest in length distributions as a sharp unimodal peak followed by a precise power-law, is exhibited both by proteins (measured in amino acids) and in software components (measured in programming language tokens).Scale independence is demonstrated in the heterogeneous behaviour of the length distributions of both proteins and software functions.Homogeneous behaviour (a Zipfian power-law distribution with a droop in the tail) is exhibited in the frequency of multiplicity in proteins and in the frequency of tokens in software.Scale independence is demonstrated in the homogeneous behaviour of both protein multiplicity and token frequency in programming languages.

These results underscore the predictable and reproducible emergence of heterogeneous and homogeneous patterns of behaviour in discrete systems, which is of particular (and underappreciated) relevance for our understanding of the structure, function, and evolution of living systems.

## Figures and Tables

**Figure 1 entropy-27-00561-f001:**
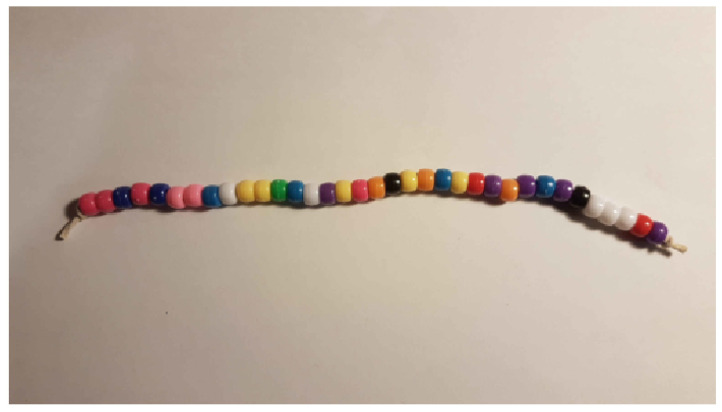
A simple string of coloured beads—or is it?

**Figure 2 entropy-27-00561-f002:**
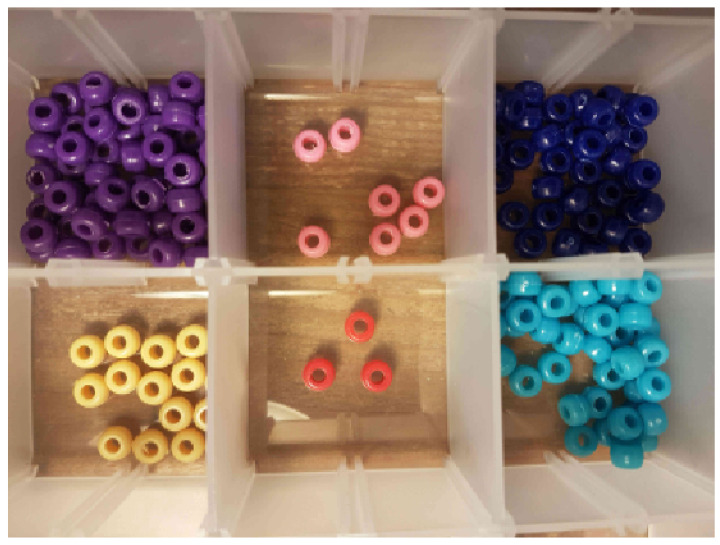
A system consisting of boxes of coloured beads. Each box contains only beads of one colour and no two boxes have the same colour. The availability and reproducibility of this figure are open access [8].

**Figure 3 entropy-27-00561-f003:**
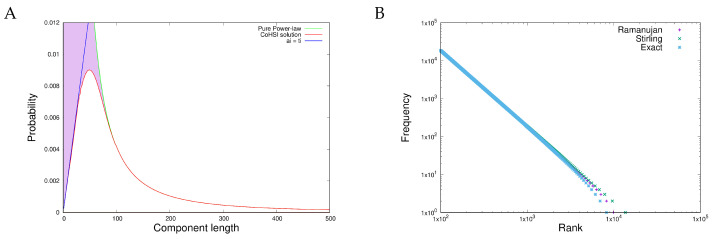
The two canonical solutions of the CoHSI Equation (Equation 16) illustrating two distinct variants of the underlying power-law behaviour. (**A**) The heterogeneous solution for string models with ordered and distinguishable beads. The green curve represents the traditional Zipf power-law solution. The red curve shows what happens for small components in this model, which lead to a sharp unimodal spike followed by an exact power-law. The blue line is a feasibility line at which the length of a particular component exactly equals its unique alphabet of colours. (**B**) The homogeneous solution for unordered bead models. In this case, the solution is a drooping-tailed version of Zipf’s eponymous law, which becomes apparent using Ramanujan’s factorial approximation as well as through exact calculation. The availability and reproducibility of this figure are open access [8].

**Figure 4 entropy-27-00561-f004:**
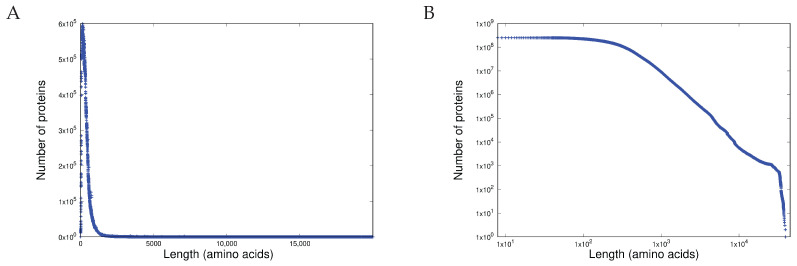
The length distribution of all approximately 251 million proteins in TrEMBL release 24-06. (**A**) Shown as a pdf linear–linear plot and (**B**) as a ccdf (cumulative complementary distribution function) and log–log plot.

**Figure 5 entropy-27-00561-f005:**
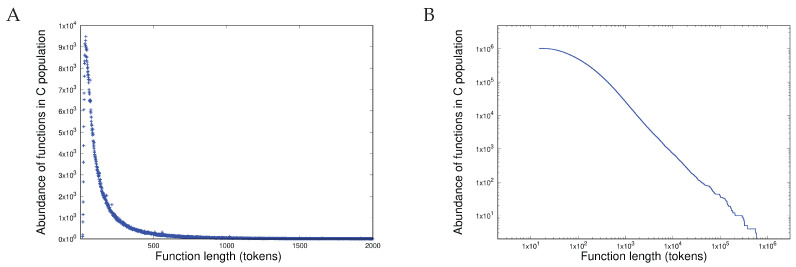
The length distribution of all 83.5 million lines of C including Linux and BSD distributions. (**A**) Shown as a pdf linear–linear plot and (**B**) shown as a ccdf (cumulative complementary distribution function) and log–log plot.

**Figure 6 entropy-27-00561-f006:**
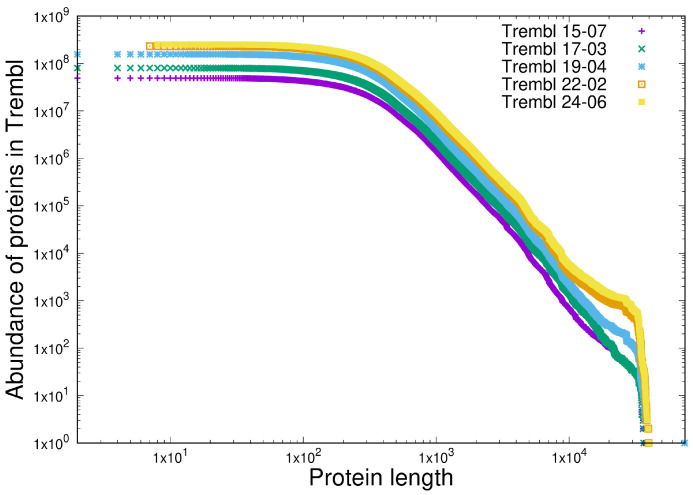
Illustrating the scale-independence of the protein length distributions. Data are shown for five TrEMBL releases spanning 10 years, in which the size of the database grew by close to a factor of 10. Protein length distributions were plotted as ccdf (cumulative complementary distribution functions) on log–log scales.

**Figure 7 entropy-27-00561-f007:**
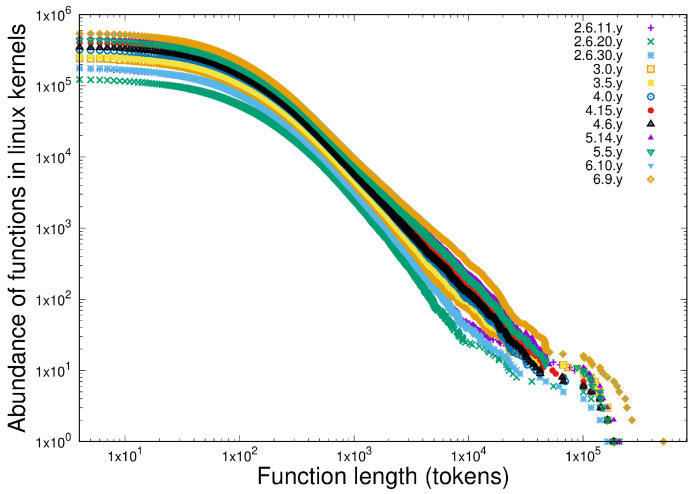
Illustrating the scale-independence of software component length distributions. Data are shown for 12 versions of the Linux kernel spanning approximately 15 years, in which the size of the kernel grew by close to a factor of 10. Software component length distributions were plotted as ccdf (cumulative complementary distribution functions) on log–log scales.

**Figure 8 entropy-27-00561-f008:**
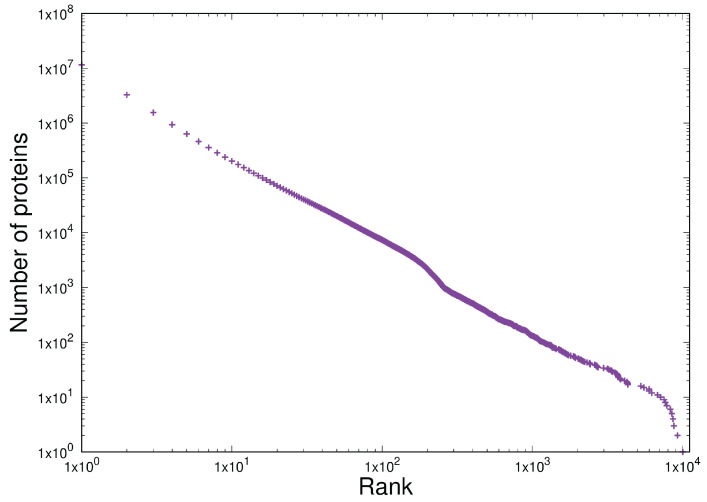
Illustrating the frequency distribution of protein multiplicity in TrEMBL release 24-06. The y-axis plots the number of proteins of a given multiplicity and the x-axis plots in rank order of multiplicity (i.e., the number of species in which a protein occurs identically). The data are shown as a ccdf on log–log scales.

**Figure 9 entropy-27-00561-f009:**
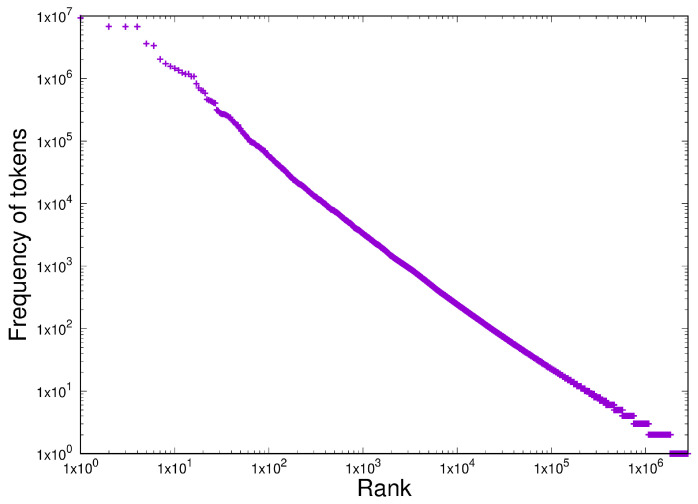
The distribution of token frequency in version 6.9 of the Linux kernel. The unique token counts were extracted and frequency *versus* rank-ordered frequency were plotted on log–log scales in the classic manner of Zipf.

**Figure 10 entropy-27-00561-f010:**
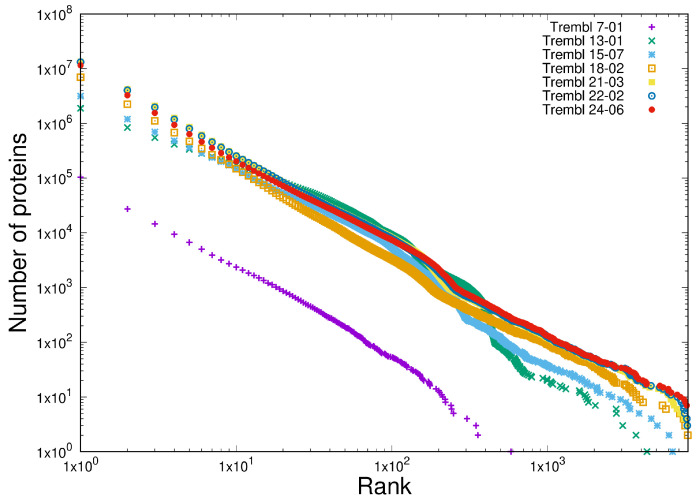
Illustrating the scale-independence in the distribution of protein multiplicity. Seven releases of the TrEMBL database are compared as the rate at which multiplicity is observed increases by a factor of 100 in the last 17 years.

**Figure 11 entropy-27-00561-f011:**
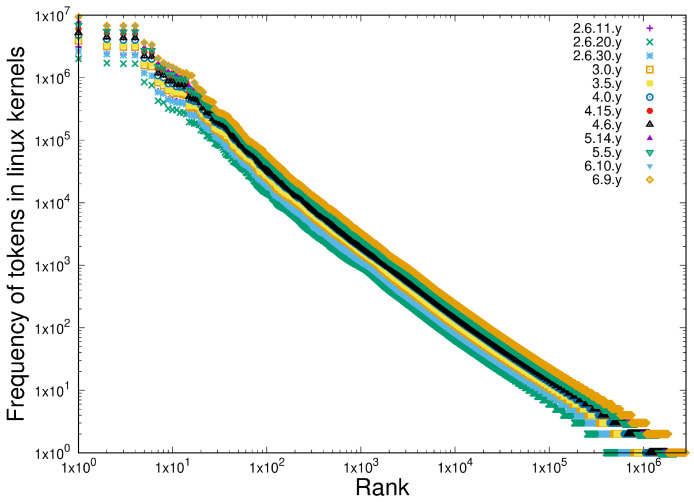
Illustrating the scale-independence in the distributions of token frequencies. Twelve releases of the Linux kernel are compared over a period of some 15 years.

**Table 1 entropy-27-00561-t001:** Statistical tests for the existence of a heterogeneous power-law in protein lengths; Figure 4B.

Test	Type	Results
R lm()	Necessary	adjusted $R^{2}$ = 0.995, *p* $\leq 2.2\times 10^{-16}$; slope, i.e., β=−3.1
Clauset–Gillespie	Sufficient	*p* = 1

**Table 2 entropy-27-00561-t002:** Statistical tests for the existence of a power-law in software function lengths, Figure 5B.

Test	Type	Results
R lm()	Necessary	adjusted $R^{2}$ = 0.99, *p* $\leq 2.2\times 10^{-16}$, β=−1.52
Clauset-Gillespie	Sufficient	*p* = 1

**Table 3 entropy-27-00561-t003:** The largest known protein in eight TrEMBL releases over time.

Release	Longest Protein	Length (aa)
7-01	BACTERIA Chlorobium chlorochromatii	36,805
13-01	BACTERIA Chlorobium chlorochromatii	36,805
15-07	BACTERIA Chlorobium chlorochromatii	36,805
17-03	BACTERIA Chlorobium chlorochromatii	36,805
18-02	EUKARYOTA Patagioenas fasciata	36,991
24-06	EUKARYOTA Hucho hucho (A0A5A9P0L4_9TELE)	45,354

**Table 4 entropy-27-00561-t004:** Statistical tests for the existence of a homogeneous power-law of the frequency of protein multiplicity in Figure 8.

Test	Type	Results
R lm()	Necessary	adjusted $R^{2}$ = 0.995, *p* $\leq 2.2\times 10^{-16}$, β=−1.33
Clauset–Gillespie	Sufficient	*p* = 1

**Table 5 entropy-27-00561-t005:** Statistical tests for the existence of a homogeneous power-law in version 6.9 of the Linux kernel.

Test	Type	Results
R lm()	Necessary	adjusted $R^{2}$ = 0.997, *p* $\leq 2.2\times 10^{-16}$, β=−1.188
Clauset-Gillespie	Sufficient	*p* = 0.1

## Data Availability

Following the recommendations of [51] all software and sources of data are openly available in a reproducibility package along with the complete means to reproduce all plots and analyses in this paper at https://leshatton.org/HattonWarr_SharedProps_Apr2025.html (accessed 22 May 2025).

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
