# Peer review of "The Origin of Shared Emergent Properties in Discrete Systems"

_entropy, 2025, doi:10.3390/e27060561_

Round 1
Reviewer 1 Report (Previous Reviewer 2)
Comments and Suggestions for Authors
This version complies with my requirements.
Author Response
Thank you.
Reviewer 2 Report (New Reviewer)
Comments and Suggestions for Authors
The article by Les Hatton and Greg Warr introduces the fascinating idea that the evolution of life is governed by the principle of constant information, called the Conservation of Hertley-Shannon Information (CoHSI). Based on this idea, the authors transform the Boltzmann distribution of statistical thermodynamics into a corresponding statistical description for information content distribution. The authors demonstrate that the corresponding statistics are being identified in databases of biological proteins as well as in software datasets. These findings are quite surprising and compelling. All findings are represented in calculated data based on the analysis of the public data sets and compared in graphical representations. Insofar, the manuscript can be regarded as a very revolutionary and unique perspective towards the evolution of life.
On the other hand, one has to consider that an amino acid sequence of a protein not only represents information, but also function. Some stretches of the peptide strand serve to form the active site, some have the function to form an anchor inside a biological membrane, some others merely connect different parts of the molecule. Is it justified to ignore these specific structural roles and to reduce all those partial sequences as mere information?
In addition, the authors argue that life could have developed without natural selection. This is a quite adventurous thought, given that natural selection and its positive effect on protein function was (and is) directly observed for microorganisms on multiple occasions.
In the theoretical section of the paper, the authors give a very detailed description on how they obtain the central distribution function for the CoHSI theory [Eq. 12]. They start with the Boltzmann distribution and argue that all basic requirements for its derivation also hold for the case of information. In this sense, the requirement for a constant total energy E also holds for the total information content I (allowing for some small fluctuation). But is this really true? In case of the total energy, this has some justification. The molecules constantly exchange energy, and the energy gain of one molecule will be the energy loss of another. I cannot see that a similar process controls the overall information. I would rather assume that a selection process could lead to an increase in information over time.
Even though I find the general outcome and the conclusions of the presented thoughts very fascinating, I have my difficulties to really believe them.
Some smaller issues should also be reconsidered by the author:
- In the text following Eq. 14, the authors state that beta is in the range of 10^21. At this point, it should be noted that, in the energy context, beta has a physical unit, which is 1/J. Hence, the magnitude of beta scales with the definition of a Joule as a unit. The comparison largely depends on the distance between energy levels, hence with the number of energy levels being populated at room temperature. Overall, the assumption of beta being much larger in the energy context than in the information context still holds for quantum systems like molecular translation or rotation (where the energy levels are very close to each other with respect to the thermal energy at room temperature), but definitely does not hold for vibration or electronic excitation.
- In section 2.3 and just before Fig. 3, the authors refer to the “heterogeneous string model”. At this point, this term is used for the first time and should be defined.
Overall, I would still support the acceptance of the manuscript. Even though I have my doubts regarding the applicability of the CoHSI theory for the evolution of life, the idea seems to be interesting enough to be published in Entropy.
Author Response
The article by Les Hatton and Greg Warr introduces the fascinating idea that the evolution of life is governed by the principle of constant information, called the Conservation of Hertley-Shannon Information (CoHSI). Based on this idea, the authors transform the Boltzmann distribution of statistical thermodynamics into a corresponding statistical description for information content distribution. The authors demonstrate that the corresponding statistics are being identified in databases of biological proteins as well as in software datasets. These findings are quite surprising and compelling. All findings are represented in calculated data based on the analysis of the public data sets and compared in graphical representations. Insofar, the manuscript can be regarded as a very revolutionary and unique perspective towards the evolution of life.
On the other hand, one has to consider that an amino acid sequence of a protein not only represents information, but also function. Some stretches of the peptide strand serve to form the active site, some have the function to form an anchor inside a biological membrane, some others merely connect different parts of the molecule. Is it justified to ignore these specific structural roles and to reduce all those partial sequences as mere information?
This is indeed a key point, thanks. In essence, we are not ignoring the physico-chemical (or other) properties of the discrete pieces, proteins or whatever, but we are separating them from those properties which devolve only from their distinguishability and which should in theory be shared by other kinds of discrete system. We therefore predict the length distribution of software components measured in programming language tokens and the length distribution of proteins measured in amino acids sharing the same shape and properties whilst the logical connections of the tokens in a software component obey a set of rules which are completely different from the physico-chemical properties of amino acids. The mathematics tells us that this should be the case but to see it acted out in such diverse systems is still a matter of wonder for us and as you describe, it remains a non-trivial point.
In addition, the authors argue that life could have developed without natural selection. This is a quite adventurous thought, given that natural selection and its positive effect on protein function was (and is) directly observed for microorganisms on multiple occasions.
This point is well taken; we agree with the reviewer that the conjecture that life could have evolved without the process of natural selection would be a bold one indeed, especially in light of the phenomena in bacteria to which the reviewer alludes. A prime example of natural selection driving evolution would be the rapid acquisition of antibiotic resistance by bacteria.
However we are not ready to make the sweeping claim that life could have evolved in the absence of natural selection. The passage in our MS to which the reviewer refers may be the following:
"This raises the interesting prospect that important properties of biological systems (exemplified here by the length and multiplicity distributions of proteins) have little if anything to do with natural selection"
We think that this statement is defensible. It derives from the point made in our response above, in which we note that there are always two sets of co-existent properties in discrete systems. The first (and generally best studied) properties are those that are specific to the system and that derive from the unique attributes of its components. The evolution of antibiotic resistance in bacteria would be a good example of such local biological properties driving evolution. In contrast the second set of properties is entirely independent of the local nature of the system components and these properties, we argue, emerge universally and probabilistically in discrete systems regardless of their provenance, meaning and mechanism.
The challenge is to reconcile these two co-existent sets of properties, the local and the universal. We agree with the generally accepted concept that evolution is driven collectively by many local processes (including neutral or random events) and we recognize the importance of natural selection. However, no matter which diverse evolutionary paths are taken by life forms, these paths will collectively and probabilistically, be "channeled" into the distributions of properties predicted by CoHSI as exemplified by the length distributions and multiplicity of proteins reported in the TrEMBL database.
In the theoretical section of the paper, the authors give a very detailed description on how they obtain the central distribution function for the CoHSI theory [Eq. 12]. They start with the Boltzmann distribution and argue that all basic requirements for its derivation also hold for the case of information. In this sense, the requirement for a constant total energy E also holds for the total information content I (allowing for some small fluctuation). But is this really true? In case of the total energy, this has some justification. The molecules constantly exchange energy, and the energy gain of one molecule will be the energy loss of another. I cannot see that a similar process controls the overall information. I would rather assume that a selection process could lead to an increase in information over time.
This may indeed be the case but we are considering the Conservation constraints in a very specific way relating to how the asymptotic distribution in Boltzmann’s theory emerges. The variational question we are asking is, (if it were energy)
Given all possible systems of the same fixed size and the same fixed total energy, what is the most likely distribution of energy amongst the various discrete pieces, (levels, molecules …)?
(whereas in the information case, we get)
Given all possible systems of the same fixed size and the same fixed total information, what is the most likely distribution of information amongst the various discrete pieces, (tokens …)?
In both cases, we are envisaging an ergodic ensemble of all possible systems with the same constraints and in both cases we could indeed imagine either further energy being injected into the system or indeed of total information increasing, but the beauty of Boltzmann’s vision is that the total size does not appear in the asymptotic distribution – the systems are scale-independent. In a gas, if you increase the total energy, you still have a Maxwell-Boltzmann distribution of velocities. In an information system, you will still have the CoHSI distributions because Boltzmann’s wonderful theory is scale-independent. This is why we see the same patterns when we allow our datasets to grow by a factor of around 10 in the paper.
Even though I find the general outcome and the conclusions of the presented thoughts very fascinating, I have my difficulties to really believe them.
We thank you for the observations. It has taken us a long time and a large number of datasets to really embrace this ourselves. The mathematics depart little from Boltzmann’s vision but the implications continue to surprise us.
We have added further text to clarify these important points.
Some smaller issues should also be reconsidered by the author:
-
In the text following Eq. 14, the authors state that beta is in the range of 10^21. At this point, it should be noted that, in the energy context, beta has a physical unit, which is 1/J. Hence, the magnitude of beta scales with the definition of a Joule as a unit. The comparison largely depends on the distance between energy levels, hence with the number of energy levels being populated at room temperature. Overall, the assumption of beta being much larger in the energy context than in the information context still holds for quantum systems like molecular translation or rotation (where the energy levels are very close to each other with respect to the thermal energy at room temperature), but definitely does not hold for vibration or electronic excitation.
A subtle point duly noted thank you. We have altered the relevant text to point out this dimensionality and its implications for the size of the energy differences relative to the total energy.
-
In section 2.3 and just before Fig. 3, the authors refer to the “heterogeneous string model”. At this point, this term is used for the first time and should be defined.
Apologies, we have changed the text appropriately. It should just be “heterogeneous case” as elsewhere.
Overall, I would still support the acceptance of the manuscript. Even though I have my doubts regarding the applicability of the CoHSI theory for the evolution of life, the idea seems to be interesting enough to be published in Entropy.
Thank you. We hope the revisions to the MS help to allay your doubts.
This manuscript is a resubmission of an earlier submission. The following is a list of the peer review reports and author responses from that submission.
Round 1
Reviewer 1 Report
Comments and Suggestions for Authors
The article try to propose an approach to seal with the Shared Emergent Properties in Discrete Systems. The Conservation of Hartley-Shannon Information (CoHSI) is introduced and the approach of Boltzmann Statistical Mechanics is applied to get the results. The addressed scientific problem is meaningful to be discussed. However, the approach is questionalble.
1, Energy is of course obeys the conservation law. But it is hard to believe that information is conservation. Actually, with the quantitative description of information by Shannon's information entropy, every special distribution gives us different information.
2, The way of getting the final Boltzamann from maxmization Ln(Ω), is based on that the equilibrium state of the thermodynamic system, is corresponds to a maximum number of microscopic states. I don't think it is the case for more open and non-equilibrium systems, such as bioligical, software systems, and others discussed in the manuscript.
3, The main results discussed in this manuscript, has already published by the authors in reference [6]. The study discussed here is nor original and novel enough to be pubilshed in Entropy.
Author Response
The article try to propose an approach to seal with the Shared Emergent Properties in Discrete Systems. The Conservation of Hartley-Shannon Information (CoHSI) is introduced and the approach of Boltzmann Statistical Mechanics is applied to get the results. The addressed scientific problem is meaningful to be discussed. However, the approach is questionalble.
1, Energy is of course obeys the conservation law. But it is hard to believe that information is conservation. Actually, with the quantitative description of information by Shannon's information entropy, every special distribution gives us different information.
We don’t claim that information is conservation. We are simply looking for the most likely of all those possible systems with the same total size and same total information content. The approach is entirely consistent with Boltzmann’s, leading to the Maxwell-Boltzmann velocity distribution which is the most likely of all possible systems with the same total size and same total energy. We have introduced more discussion of this which we hope makes the point clearly.
2, The way of getting the final Boltzamann from maxmization Ln(Ω), is based on that the equilibrium state of the thermodynamic system, is corresponds to a maximum number of microscopic states. I don't think it is the case for more open and non-equilibrium systems, such as bioligical, software systems, and others discussed in the manuscript.
We are looking for shared emergent properties. The only common denominator which could link systems as diverse as proteins and computer software is if the only assumption made is that tokens are distinguishable but have no intrinsic meaning. The only assumption we make above the normal assumptions of statistical mechanics is that all symbols are therefore equally likely as Hartley originally proposed. When we do this and derive a theory predicting what the most likely state will be given the absence of intrinsic meaning, we derive two distributions which we call the CoHSI distributions. We then demonstrate their presence in both software and proteins, strongly supporting our hypothesis that discrete systems of completely different provenance share global emergent properties. More written material has been added to help round this out.
3, The main results discussed in this manuscript, has already published by the authors in reference [6]. The study discussed here is nor original and novel enough to be pubilshed in Entropy.
We have not so far published a head-on comparison of the heterogeneous and homogeneous CoHSI distributions for two completely disparate systems with the object of showing that all discrete systems have shared emergent properties. We hope this makes it clear.
Thank you for your feedback.
Reviewer 2 Report
Comments and Suggestions for Authors
This is an interesting paper. The problem is that it is essentially equivalent to your Royal Society Open Science 2019 publication, which you often quote in the text. Duplicating publications is not acceptable, and making a paper dependent on another paper is bad practice. Scientific papers should be largely self-contained, whereas in this case I had to read your 2019 paper as well.
So the first thing to do is that - although you can still cite your 2019 paper as a reference to deepen certain topics - this paper becomes self-contained. If you think that moving portions of your 2019 paper into this paper would destroy the flow of arguments, you may put the additional contents in an appendix.
Obviously, solving dependence will make duplication even more evident. These two papers will become even more similar to one another, so now similarity is the main problem to address.
One common strategy for publishing similar papers is that they are applied to different data. This is the strategy I am going to suggest henceforth.
Both in this and in your 2019 paper you claim that your findings are independent of computer language, but you only carried out one empirical analysis on functions in C. In order to make such a bold claim it is absolutely necessary that you test at least one other language. For instance, you could check your claim with objects in Java or Python besides functions in C.
Alternatively, you could avoid this cumbersome additional investigation by making a more humble claim in this paper, namely, that your findings are limited to the C language. But you already made the bold claim concerning all computer languages in your 2019 paper, so now you are exposed to criticisms anyway. With this paper, you have an opportunity to defend yourself from criticisms by highlighting the same regularity in at least one other computer language.
One important content of your paper is that, given one and the same data, you obtain different statistics depending on whether you observe sequences of tokens, or frequencies of their occurrence. However, you claim, statistics based on sequences are similar to one another across datasets, and statistics based on frequencies are similar to one another across datasets. Specifically, here are your claims:
D1: You show the similarity of statistics on sequences of amino acids to the statistics on sequences of software instructions.
D3: You show the similarity of statistics on frequencies of amino acids to the statistics on frequencies of words in literary texts.
Switching database from software to literary text doesn't make a good impression. Since you claim generality, you should select two databases and use them consistently. Either (proteins,software) in both D1 and D3, or (proteins,texts) in both D1 and D3. Or, even better, the three databases (proteins,software,texts) in both D1 and D3.
Is there a reason for your choice? I can figure out that identifying exact sequences of words may yield too few repetitions of each sentence, but measuring the frequency of software instructions shouldn't be difficult. That is, applying the datasets (proteins,software) to both D1 and D3 shouldn't be a problem.
If you don't apply the same datasets to D1 and D3, readers may think that you are hiding a case where you failed in order to highlight regularities that only show up in other cases. Hence it is very important that, once you have introduced a dataset, you explore it consistently.
In D2, you illustrate scale invariance of protein length. In D4, you illustrate scale invariance of protein multiplicity. Again, can you find similar results in the other databases? The theoretical part of your paper is about seeking universal regularities, D2 and D4 shouldn't be limited to one single database after you have introduced three.
In the case of texts, identifying production time is not difficult (say, the Bible comes before Shakespeare). In the case of software, you should be able to find older versions. I mean, it shouldn't be difficult to generate databases that grow with time, just like the proteins database. In your 2019 paper you cite a IEEE paper by one of you, where you apparently found out a similar regularity in "software systems" (but I don't know what you mean).
I am aware that carrying out all of the above analyses is a big work, and I understand that this is a subject for a long series of papers. Here is a list of possible improvements, in steps of increasing work. I am just requesting that you carry out the first of them. If you want, you can carry out (2), or (3), or (4). Or. postpone those analyses to future papers.
1) Carry out D3 on C software. The outcome is: D1 and D3 on proteins and C software. You must specify in the abstract that you focus on C only, and only a portion of properties. You can skip D2 and D4 altogether, leaving them for a subsequent paper.
2) Carry out D3, D2, D4 on C software. The outcome is: D1, D2, D3 and D4 on proteins and C software. You must specify in the abstract that you focus on C only.
2) Carry out D1, D2, D3, D4 on C and Java/Python. The outcome is: D1, D2, D3 and D4 on proteins and software in at least two languages. You keep the bold claims you made in the abstract.
2) Carry out D1, D2, D3, D4 on C and Java/Python, as well as on literary texts. The outcome is: D1, D2, D3 and D4 on proteins, software in at least two languages, and in literary texts. The claims you can make in the abstract become even bolder.
LESS FUNDAMENTAL, BUT NOT MINOR ISSUES
Conclusions are three lines long. This is very irritating. I'm doing my job, please do yours.
Lines 75-79: "For a closed system, changes in S are non-negative and it is a mistake to think that life (...) can alter this fact." You quoted a 1918 source in German. It may be fine, but it makes better sense that you quote someone who criticised Schroedinger. Here is one such reference, from this journal:
- Jefferey, K., Pollack, R. and Rovelli, C. "On the Statistical Mechanics of Life: Schroedinger Revisited." Entropy 2019, 21, 1211.
Zipf means a power law with exponent 1. Since you found exponents of diverse values, you should speak of power laws rather than Zipf.
Moreover, in the analysis of literary texts you found an exponent 0.78 instead of 1. Why?
Your usage of the Rubin's vase is inappropriate, because interpreting that image as either a vase or two faces implies switching between different meanings of one and the same information. With the Rubin example you implicitly suggest that the meaning of the string of coloured beads can switch between "wrist band" and "protein." But you say that you are not concerned with meaning, you just want to compare sequences to frequencies of tokens.
Please drop Rubin's vase altogether. I would rather suggest to combine Figures 1 and 2. Delete the Rubin thing, put Figure 2 in its place.
Author Response
This is an interesting paper. The problem is that it is essentially equivalent to your Royal
Society Open Science 2019 publication, which you often quote in the text. Duplicating
publications is not acceptable, and making a paper dependent on another paper is bad
practice. Scientific papers should be largely self-contained, whereas in this case I had to
read your 2019 paper as well.
Apologies. We fell between two hurdles here. The original invitation was based around the 2019 paper. Given the time scale, we thought to extend this with a discussion of beta in the two formulations followed by similar analyses (although all the datasets are different) to the 2019 paper, so we accept your criticism, although one of the difficulties we have faced on a number of occasions is trying to make papers self-contained as the 2019 paper is quite a long one.
So the first thing to do is that - although you can still cite your 2019 paper as a reference to
deepen certain topics - this paper becomes self-contained. If you think that moving portions
of your 2019 paper into this paper would destroy the flow of arguments, you may put the
additional contents in an appendix.
We have taken your advice and added an Appendix which we hope acts as a sufficient bridge between the original RSOS paper (in which the arguments are rather lengthy) and this one so that it becomes self-contained.
Obviously, solving dependence will make duplication even more evident. These two papers
will become even more similar to one another, so now similarity is the main problem to
address.
One common strategy for publishing similar papers is that they are applied to different data.
This is the strategy I am going to suggest henceforth.
Both in this and in your 2019 paper you claim that your findings are independent of
computer language, but you only carried out one empirical analysis on functions in C. In
order to make such a bold claim it is absolutely necessary that you test at least one other
language. For instance, you could check your claim with objects in Java or Python besides
functions in C.
The 2019 paper focused on C for reasons given below but referenced Hatton and Warr (2017) “Information Theory and the Length Distribution of all Discrete Systems”, arXiv. 2017 Sep;http://arxiv.org/pdf/1709.01712 [q-bio.OT] which does consider multiple programming languages in depth including Fortran, Ada, Java, C++ and a couple of others. This is a very long paper so we could not easily assimilate it in the current MS. In general dealing with multiple languages is a logistically problematic area for several reasons.
- The bulk of the open source software available for analysis is still in C, (something like 60-70% depending on how you count it – Hatton, Spinellis and van Genuchten (2017), doi 10.1002/smr.1847). You can forget closed source, its basically unobtainable.
- Introducing a new programming language requires writing enough of a front-end of a compiler to perform reliable component and token extraction. This is not trivial, particularly for languages which blur lexical and syntactic analysis like C++. It took 18 months to write parsers for C, C++, Fortran, Ada, Java, TclTk and perl of sufficient quality to do this. Of these the C parser is the most accurate and was consistent with FIPS 160. You make an excellent point though and we have included a discussion of this in the revised MS for the interested reader.
Alternatively, you could avoid this cumbersome additional investigation by making a more
humble claim in this paper, namely, that your findings are limited to the C language. But
you already made the bold claim concerning all computer languages in your 2019 paper, so
now you are exposed to criticisms anyway. With this paper, you have an opportunity to
defend yourself from criticisms by highlighting the same regularity in at least one other
computer language.
By referencing both the 2019 paper and the 2017 paper above explicitly, along with supporting text in the revised MS, we hope we have suitably defended ourselves.
One important content of your paper is that, given one and the same data, you obtain
different statistics depending on whether you observe sequences of tokens, or frequencies
of their occurrence. However, you claim, statistics based on sequences are similar to one
another across datasets, and statistics based on frequencies are similar to one another across
datasets. Specifically, here are your claims:
D1: You show the similarity of statistics on sequences of amino acids to the statistics on
sequences of software instructions.
D3: You show the similarity of statistics on frequencies of amino acids to the statistics on
frequencies of words in literary texts.
Switching database from software to literary text doesn't make a good impression. Since
you claim generality, you should select two databases and use them consistently. Either
(proteins,software) in both D1 and D3, or (proteins,texts) in both D1 and D3. Or, even
better, the three databases (proteins,software,texts) in both D1 and D3.
Is there a reason for your choice? I can figure out that identifying exact sequences of words
may yield too few repetitions of each sentence, but measuring the frequency of software
instructions shouldn't be difficult. That is, applying the datasets (proteins,software) to both
D1 and D3 shouldn't be a problem.
If you don't apply the same datasets to D1 and D3, readers may think that you are hiding a
case where you failed in order to highlight regularities that only show up in other cases.
Hence it is very important that, once you have introduced a dataset, you explore it
consistently.
In D2, you illustrate scale invariance of protein length. In D4, you illustrate scale invariance
of protein multiplicity. Again, can you find similar results in the other databases? The
theoretical part of your paper is about seeking universal regularities, D2 and D4 shouldn't
be limited to one single database after you have introduced three.
In the case of texts, identifying production time is not difficult (say, the Bible comes before
Shakespeare). In the case of software, you should be able to find older versions. I mean, it
shouldn't be difficult to generate databases that grow with time, just like the proteins
database. In your 2019 paper you cite a IEEE paper by one of you, where you apparently
found out a similar regularity in "software systems" (but I don't know what you mean).
I am aware that carrying out all of the above analyses is a big work, and I understand that
this is a subject for a long series of papers. Here is a list of possible improvements, in steps
of increasing work. I am just requesting that you carry out the first of them. If you want,
you can carry out (2), or (3), or (4). Or. postpone those analyses to future papers.
1) Carry out D3 on C software. The outcome is: D1 and D3 on proteins and C software.
You must specify in the abstract that you focus on C only, and only a portion of properties.
You can skip D2 and D4 altogether, leaving them for a subsequent paper.
2) Carry out D3, D2, D4 on C software. The outcome is: D1, D2, D3 and D4 on proteins
and C software. You must specify in the abstract that you focus on C only.
3) Carry out D1, D2, D3, D4 on C and Java/Python. The outcome is: D1, D2, D3 and D4 on
proteins and software in at least two languages. You keep the bold claims you made in the
abstract.
4) Carry out D1, D2, D3, D4 on C and Java/Python, as well as on literary texts. The
outcome is: D1, D2, D3 and D4 on proteins, software in at least two languages, and in
literary texts. The claims you can make in the abstract become even bolder.
Thank you for this analysis. We have taken 2) for a variety of reasons which we ought to mention. 1) is more restrictive than what we already have. Accumulating sufficient Java/Python indexed by date is infeasible in the time available and possibly infeasible generally at the moment. The source code extracted in Hatton, Spinellis and van Genuchten (2017), doi 10.1002/smr.1847 took a couple of months of running a robot. The majority of the 200 million or so lines of source code pulled were C. This rules out D2 and D4 on Java/Python and probably other programming languages. In turn this rules out options 3) and 4). It would be interesting to do but source code archival is mostly a complete shambles compared with biological datasets such as those held by Uniprot. It could be one of several source code control systems, often poorly indexed, with frequent duplications – there is little if any curation. The linux kernel is a shining but relatively rare example of how to do it properly.
D2, D3 and D4 on C software are feasible. We thought that probably the most reliable solution is to compare source downloads of the Linux kernel for heterogeneous and homogeneous behaviour. This is both large and curated carefully. Downloading just involves git cloning the stable branch of the kernel archive – around 5GB compressed. Each checked out version can then be extracted, parsed, and analysed. This we have done for 12 versions spread across the last few years up to the present day, an interesting project in itself.
We have also taken the opportunity to reference prior work demonstrating those results we have for multiple programming languages.
LESS FUNDAMENTAL, BUT NOT MINOR ISSUES
Conclusions are three lines long. This is very irritating. I'm doing my job, please do yours.
Duly extended. Apologies.
Lines 75-79: "For a closed system, changes in S are non-negative and it is a mistake to
think that life (...) can alter this fact." You quoted a 1918 source in German. It may be fine,
but it makes better sense that you quote someone who criticised Schroedinger. Here is one
such reference, from this journal:
- Jefferey, K., Pollack, R. and Rovelli, C. "On the Statistical Mechanics of Life:
Schroedinger Revisited." Entropy 2019, 21, 1211.
Certainly worth quoting, thank you which we have now done, but we would have thought breaching something as fundamental as the second law of thermodynamics almost doesn’t need a reference.
Zipf means a power law with exponent 1. Since you found exponents of diverse values, you
should speak of power laws rather than Zipf.
Moreover, in the analysis of literary texts you found an exponent 0.78 instead of 1. Why?
These are interesting points, thanks. Generally speaking in the literature, Zipf has become synonymous with power-law so “Zipfian” behaviour is often used to describe power-law behaviour in biological texts for example independently of the exponent. Its original association with a unit gradient power-law is almost lost in time and in practice as we found here, it quite often departs from unity. We have developed this more in the text to respond to this. Its also worth noting again that neither CoHSI distribution is strict power-law. Both have power-law behaviour over significant parts of their support but each differs in its own way, (homogeneous naturally droops and heterogeneous naturally spikes).
We haven’t explored this further in the text as we are sticking to option 2) as you described, although it is an interesting subject in itself for which we have quite a lot of data.
Your usage of the Rubin's vase is inappropriate, because interpreting that image as either a
vase or two faces implies switching between different meanings of one and the same
information. With the Rubin example you implicitly suggest that the meaning of the string
of coloured beads can switch between "wrist band" and "protein." But you say that you are
not concerned with meaning, you just want to compare sequences to frequencies of tokens.
Please drop Rubin's vase altogether. I would rather suggest to combine Figures 1 and 2.
Delete the Rubin thing, put Figure 2 in its place.
We see your point. We clearly didn’t explain this too well. We were trying to say that when you free your perception of specific meaning then a string of beads can either be a wrist band or a protein. All that remains is distinguishability and consequently, an information content which is purely probabilistic since all symbols are equally likely. This Hartley information content yields a distribution based purely on the alphabet of distinguishable choices without any regard for what they might mean. We were using Rubin purely to illustrate that calculating probabilities based on distinguishability alone compared with for example, thinking about the diverse and intricate physico-chemical properties of amino acids, is a psychological leap comparable to switching between two different perceptions of the same image. As a matter of interest, we have observed that biologists often find it difficult to make this switch because of the all-consuming complexity of interactions in the genome or protein, which is why we thought it worth discussing. However if we are not getting our point across, we are clearly not improving the MS.
Your suggestion of deleting Mr. Rubin sounds much simpler given the space we have, thanks, so we have adjusted the MS appropriately.
Finally, many thanks for these various suggestions, you have clearly spent a lot of time on this and we are very grateful. We hope the MS reads much more fluently now.
Round 2
Reviewer 1 Report
Comments and Suggestions for Authors
I can not agree with the idea of the research. So I am sorry that I can not suggest its publication in Entropy.
Author Response
"I can not agree with the idea of the research. So I am sorry that I can not suggest its publication in Entropy."
In effect you are rejecting not only all of the new work we have prepared for Entropy but also the previously peer-reviewed work in a Royal Society Open Science paper. To find fault with any of this work, you must either demonstrate that the (peer-reviewed) mathematics is incorrect or that the (third party open access) datasets are inappropriate or duplicated, or that there are errors in the software used to analyse the data (we provide the full means to verify these results). You have done none of these. You have simply said you don’t like it. You are of course entitled to your opinion but with respect, that does not constitute a credible review of a scientific paper.
Reviewer 2 Report
Comments and Suggestions for Authors
Thank you for considering my suggestions. Good luck with the next publications!
Author Response
"Thank you for considering my suggestions. Good luck with the next publications!"
Thank you for your careful feedback and the hard work you obviously put into it. It really was very useful.